# Plant Growth Enhancement using Rhizospheric Halotolerant Phosphate Solubilizing Bacterium *Bacillus licheniformis QA1* and *Enterobacter asburiae QF11* Isolated from *Chenopodium quinoa* Willd

**DOI:** 10.3390/microorganisms8060948

**Published:** 2020-06-24

**Authors:** Ismail Mahdi, Nidal Fahsi, Mohamed Hafidi, Abdelmounaaim Allaoui, Latefa Biskri

**Affiliations:** 1Medical Application Interface Center (CIAM), Mohammed VI Polytechnic University (UM6P), 43150 Benguérir, Morocco; Ismail.mahdi@um6p.ma (I.M.); nidal.fahsi@um6p.ma (N.F.); 2Laboratory of Microbial Biotechnologies, Agrosciences and Environment (BioMAgE), Faculty of Sciences Semlalia, Cadi Ayyad University, 40000 Marrakesh, Morocco; mohamed.hafidi@um6p.ma; 3Laboratory of Genetic, Neuroendocrinology and Biotechnology, Faculty of Sciences, Ibn Tofail University, 14000 Kénitra, Morocco; 4Microbiome Team and African genome center (AGC), AgrobioSciences department (AgBS), Mohammed VI Polytechnic University (UM6P), 43150 Benguérir, Morocco; abdelmounaaim.allaoui@um6p.ma; 5Laboratory of Molecular Microbiology, CIPEM (Coalition Center, for Innovation, and Prevention of Epidemies in Morocco) Mohammed VI Polytechnic University (UM6P), 43150 Benguérir, Morocco

**Keywords:** phosphate solubilizing bacteria, *Chenopodium quinoa*, salt stress, IAA, seedling growth, germination, plant growth promotion

## Abstract

Plant growth-promoting rhizobacteria represent a promising solution to enhancing agricultural productivity. Here, we screened phosphate solubilizing bacteria from the rhizospheric soil of *Chenopodium quinoa Willd* and assessed their plant-growth promoting rhizobacteria (PGPR) properties including production of indole-3-acetic acid (IAA), siderophores, hydrogen cyanide (HCN), ammonia and extracellular enzymes. We also investigated their tolerance to salt stress and their capacity to form biofilms. Two isolated strains, named QA1 and QF11, solubilized phosphate up to 346 mg/L, produced IAA up to 795.31 µg/mL, and tolerated up to 2 M NaCl in vitro. 16S rRNA and Cpn60 gene sequencing revealed that QA1 and QF11 belong to the genus *Bacillus licheniformis* and *Enterobacter asburiae*, respectively. In vivo, early plant growth potential showed that quinoa seeds inoculated either with QA1 or QF11 displayed higher germination rates and increased seedling growth. Under saline irrigation conditions, QA1 enhanced plant development/growth. Inoculation with QA1 increased leaf chlorophyll content index, enhanced P and K^+^ uptake and decreased plant Na^+^ uptake. Likewise, plants inoculated with QF11 strain accumulated more K^+^ and had reduced Na^+^ content. Collectively, our findings support the use of QA1 and QF11 as potential biofertilizers.

## 1. Introduction

Soil microbiome, vegetation, and soil fauna are strongly affected by climate change which causes progressive increased land salinization and desertification [1]. Salinity is a limiting factor for crop production in semi-arid and arid soils, mainly due to the accumulation of sodium chloride (NaCl) [2]. Salt stress changes the physicochemical characteristics of soils and disrupts the growth of both plants and soil microflora [3]. Thus, there is a selection pressure for stress-adapted and extremophilic plants [1]. The use of salt-tolerant plants and microorganisms is a promising strategy to alleviate salt-induced effects on crops. 

Through root exudates, plants secrete organic compounds having selective effects on associated microbes. They attract beneficial bacteria, known as plant growth promoting bacteria (PGPB) which subsequently boost agricultural productivity. Indeed, the use of PGPB as biofertilizers improves the availability of phosphorus and nutrients to crops, ameliorates soil structure and promotes the health and the fertility of arable soils [4,5,6].

Phosphorus (P) is one of the undeniable elements in plant nutrition alongside nitrogen (N) and potassium (K). Therefore, acquisition of sufficient P concentration is a critical process for plant growth [7]. Soil contains a substantial reserve of P in both inorganic and organic forms. Indeed, it has been estimated that accumulated P in arable lands would be sufficient to sustain maximum crop production worldwide for about 100 years [8]. However, plants are only able to assimilate P in its soluble forms i.e., orthophosphates (HPO_4_^2−^, H_2_PO_4_^−^), which represent only 0.1 to 0.5% of the total soil’s P [9]. P is found in soils mainly in mineral complexes such as tricalcium phosphate (Ca_3_ (PO_4_)_2_), iron phosphate (FePO_4_), aluminum phosphate (AlPO_4_) [10], and in phytate form which constitutes the majority of soil organic P [11]. To tackle this issue, P-fertilizers are generally recommended. Nevertheless, according to their forms and methods of application, more than 80% of P in applied fertilizers is instantly fixed via immobilization, sorption or precipitation reactions leading to low P uptake and reduced fertilization efficiency [9]. Soil PGPB constitute very important components in the global cycling of soil nutrients, including P [12]. They are involved in maintaining the fertility of soils, optimizing plant nutrition and strengthening the root system [13,14]. Natural solubilization of unavailable P is an enthralling phenomenon exhibited by various soil microorganisms, commonly known as phosphate solubilizing microorganisms (PSM). Among these P-solubilizers, bacteria (PSB) represent the predominant group as compared to other microbes [15]. Indeed, diverse bacterial species, particularly rhizosphere colonizing bacteria, have the ability to mineralize organic P and solubilize inorganic P compounds such as hydroxyapatite, rock phosphate and tricalcium phosphate [16]. Several strategies, mainly genetic engineering and the use of PGPB [17], have been used to attempt to reduce the damaging effect of salt stress on plant growth. 

Besides their role in making P available to plants, PSB are useful in mitigating abiotic stresses such as salinity and drought via the release of different phytohormones, antioxidant metabolites and enzymes as well as regulation of ion uptake [18]. It has been reported that salt stress will reduce arable lands by 50% in 2050 [19]. This is mainly due to the high level of soluble salt ions in the soil such as sodium (Na^+^), bicarbonate (HCO3^−^), chloride (Cl^−^), calcium (Ca^2+^) and carbonate (CO_3_^2−^). Indeed, the high level of NaCl is the most toxic for plants [2]. Saline soils are defined as soils in which the exchangeable Na^+^ rate is 15% and the electrical conductivity of the saturated extract in the soil surpasses 40 mM NaCl [3]. In fact, the rate of saline soil is increasing because of multiple factors such as saline irrigation, ground water salinity and low precipitation [20]. In terms of abiotic stresses, salinity is the most devastating stress affecting soil quality, plants and microorganism bio-diversity [18]. 

Using halotolerant PSB to increase soluble P concentrations in P-poor agricultural soils and to mitigate the harmful effects of salinity has considerable potential for modern agriculture. Quinoa (Chenopodium quinoa Willd), a herbaceous plant species belonging to the Amaranthaceae family, is renowned for its ability to withstand drought and salinity [21] and for its remarkable nutritional value [22]. Quinoa has been recently introduced in Morocco for its revenue-generating potential, its adaptability to different soil and climatic conditions and for its potential to improve cropping systems [23]. 

The characterization of halotolerant PSB-colonizing quinoa plant rhizospheres and their effects on plant growth have not yet been reported in Morocco. In the present study, we isolated two bacterial strains from the rhizosphere of *Chenopodium quinoa* endowed with P solubilization and additional plant-growth promoting rhizobacteria (PGPR) properties. We also investigated the effect of their inoculation on quinoa grown under salinity stress. Together, our data support the potential use of QA1 and QF11 strains as biofertilizers.

## 2. Materials and Methods 

### 2.1. Soil Sampling 

Twelve rhizospheric soil samples of 3-month-old quinoa plants were collected from quinoa fields of an experimental farm (32.219731E, −7.892268N) at Mohammed VI Polytechnic University-Benguerir, Morocco. Sampling was carried out in June 2018, peak growing season for quinoa. The soil samples were taken within 20–30 cm of quinoa plants. Samples including roots and soil aggregates weighing approximately 50 g were collected from each site, placed individually in sterile plastic bags, stored at 4 °C and immediately transported to the laboratory. All samples were stored at 4 °C until use [24]. 

### 2.2. Isolation and Screening of Phosphate Solubilizing Bacteria on Plates

One g of soil from each sample was aseptically transferred to a tube containing 9 mL of sterile distilled water and then shaken for 30 min. Afterwards, a series of decimal dilutions (10^−1^ to 10^−6^) of the soil suspensions were carried out in Eppendorf tubes. To isolate potential PSB, 100 μL of each dilution was plated on TSA (EMD Millipore, Berlin, Germany) nutrient agar following the standard spread plating technique. A total of 79 bacterial isolates exhibiting different morphological aspects were spot inoculated on NBRIP (National Botanical Research Institute’s phosphate) agar medium plates [25] consisting of (g/L) dextrose 10; hydroxyapatite 5 (*purum p.a.*, *≥ 90% (as Ca^3^(PO4)_2_,KT*); ammonium sulphate 0.5; potassium chloride 0.2; sodium chloride 0.2; magnesium sulphate 0.1; ferrous sulphate trace; manganese sulphate trace; and agar 15. The pH was adjusted to 6.75 ± 0.25 before autoclaving [16]. Plates were incubated at 30 °C and checked daily over 7 days for the appearance of transparent halos indicating P-solubilizing ability. Colonies showing discrete halo zones were purified on the same medium and stored at −80 °C in cryotubes using 10% dimethyl sulfoxide (DMSO) as a cryoprotective agent.

### 2.3. Quantitative Assay of Phosphate Solubilization in Liquid Medium

The plate assay is a relative efficiency test and is not a precise method to identify whether or not a strain is a P solubilizer [26]. Thus, P solubilization activity was then quantified using hydroxyapatite as a sole source of insoluble P in NBRIP broth medium [25]. The isolates were individually grown overnight in TSB broth medium and the optical density (OD_600nm_) was adjusted to 0.8. The bacterial suspension (100 μL) was inoculated in 250 mL Erlenmeyer flasks containing 50 mL of NBRIP broth. The resulting media was incubated for 5 days at 30 °C under 150 rpm shaking and centrifuged at 12,000 rpm for 10 min [16]. The supernatants were filtered through 0.22 μm sterile syringe filters to remove insoluble materials [14]. The cell-free supernatants were diluted (1/50) and used to colorimetrically measure the soluble P content using a Continuous Flow Analyzer (SKALAR SAN++ SYSTEM). Uninoculated NBRIP medium served as a control. The final pH of the supernatants was also recorded. The experiment was performed in triplicate. 

### 2.4. DNA Amplification and Sequences Analysis 

The taxonomic identification of selected PSB was carried out using the 16S rRNA and the chaperonin-60 (Cpn60) genes sequencing. Primers pA (5′-AGAGTTTGATCCTGG CTCAG-3′) and 926R_Quince (5′-CCGYCAATTYMTTTRAGTTT-3′) [27] were used to amplify the 16S rRNA gene, while degenerated primers H279 (5′-GAIIIIGCIGGIGAYGGIACIACIAC-3′) and H280 (5′-YKIYKITCI CCRAAICCIGGIGCYTT-3′) were used for Cpn60 gene amplification [28]. All PCR reactions were carried out in 50 μL final volume containing 23 μL DNAase free water, 25 μL MyTaq Mix (Thermo Fisher, Casablanca, Morocco), 1 μL of forward and reverse primers at 20 µM final concentration, and 1 μL of fresh overnight bacterial cultures as DNA matrix. The PCR cycling for both *16S rRNA* and *Cpn60* was performed, using the VWR^®^ thermal cycler, as follows: initial denaturation at 94 °C for 5 min, followed by 35 cycles of denaturation at 94 °C for 30 s, annealing at 54 °C for 30 s and elongation at 72 °C for 1 min, with the final extension programmed at 72 °C for 10 min. Used primers for 16S rRNA gene target variable regions V1–V5. The PCR products (~1000 bp for *16S rRNA* sequences, and ~600 bp *cpn60* amplicons) were commercially sequenced. Generated DNA sequences were analyzed using SILVA program [29] (for 16S rRNA sequences) and alignment search using BLAST tool on NCBI (for 16S rRNA and Cpn60 sequences) database [30]. The 16S rRNA sequences were deposited to NCBI GenBank and the accession numbers were provided. The phylogenetic tree was built using the neighbor-joining method using UGENE software [31].

### 2.5. In Vitro Screening for PGP Activities

#### 2.5.1. Indole-3-Acetic Acid (IAA) Production Assay

Quantification of IAA (indole-3-acetic acid) produced by PSB strains was estimated by growing them in TSB broth amended with 0.1% L-tryptophan as a precursor of IAA [32]. Selected PSB were cultured in 50 mL of prepared medium and incubated at 28 ± 2 °C and 150 rpm shaking for 7 days [33]. The bacterial cultures were next centrifuged at 12,000 rpm for 10 min at 4 °C and the supernatants were filtered through 0.22 μm sterile syringe filters. Two mL of *Van Urk Salkowski* reagent consisting of 1 mL of 0.5 M FeCl_3_ and 50 mL of 35% HClO_4_ was mixed with 1 mL of each filtrate. Following incubation in a dark space for 30 min at room temperature, the development of a pinkish color indicated the production of IAA. Absorbance was then measured at 535 nm. The concentrations of IAA produced were estimated according to a standard curve using pure IAA (Sigma Aldrich, Overijse, Belgium) for concentrations in 0–100 µg/mL range.

#### 2.5.2. Bacterial Salinity and Heat Stress Monitoring

To assess the salinity tolerance of selected PSB, each isolate was streaked on TSA plates supplemented with different NaCl concentrations ranging from 0 to 2400 mM and incubated at 30 °C for 48 h. Bacteria were further tested in TSB broth using 48-well microtiter microplates to determine the maximal growth and minimal inhibitory salt concentrations. Briefly, 2 μL of each bacterial culture (OD_600nm_ = 0.8) was inoculated in 500 μL TSB supplemented with various NaCl concentrations (0 to 2700 mM) and incubated at 30 °C under shaking of 150 rpm. Growth patterns after 48 h incubation were measured at 600 nm using the VICTOR Nivo^TM^ Multimode Plate Reader (Perkin Elmer, Casablanca, Morocco). [34]. Selected PSB were also examined for their ability to tolerate heat stress. In brief, we streaked each bacterium on TSA medium and the plates were incubated at different temperatures ranging from 30 to 60 °C. Post 24 h of incubation, thermotolerance were determined by observing bacterial growth on plates [35].

#### 2.5.3. Biofilm Formation Assay

Biofilm formation was assessed using the colorimetric assay [36]. Fresh overnight culture of each bacterium was diluted to 1/100 in TSB broth and 200 μL of each bacterial suspension (OD_600nm_ = 0.8) was inoculated in triplicate into a 48-well microtiter microplate. Uninoculated media was used as a negative control. The microplate was incubated at 30 °C for 24 h. Next, the supernatants were aspirated using *VACUSIP* system and the bacterial pellets were washed three times with 200 μL of the phosphate-buffered saline (PBS) to remove planktonic bacteria. Biofilm formation was monitored by adding of 2% crystal violet for 15 min at room temperature. The excess dye was removed by washing with distilled water. The bacterial biofilm was solubilized using 200 μL of 95% ethanol and the OD_600nm_ was measured using the VICTOR Nivo^TM^ Multimode Plate Reader. The OD values were taken as an index of biofilm formation.

#### 2.5.4. Siderophores Production Assay 

Qualitative assessment of siderophore-producing capacity of selected PSB was performed using agar CAS (Chrome Azurol S) assay [37]. Four solutions were used: Fe-CAS indicator solution (1), buffer solution (2), mixing salt solution (3) and casamino acid solution (4). To prepare solution (1), 0.06 g CAS was dissolved in 50 mL distilled water and mixed with 10 mL iron solution (1 mM FeCl_3_.6H_2_O, 10 mM HCl). Under shaking condition, this solution was added to 0.073 g HDTMA (hexadecyltrimethylammonium bromide) dissolved in 40 mL distilled water. The blue solution was autoclaved and cooled to 55 °C. To prepare solution (2), 32.24 g PIPES (Pipérazine-N, N’-bis (2-éthanesulfonique)) was dissolved in 7.5% of solution (3) containing (in 100 mL distilled water) 3 g KH_2_PO_4_, 5 g NaCl and 10 g NH_4_Cl. The pH of the PIPES solution was adjusted to 6.8. Solution (2) was autoclaved after adding 15 g agar to solution (4), 3 g of casamino acid was dissolved in 27 mL of distilled water and sterilized using 0.22 µm filters. The four solutions were mixed and poured on plates. After solidifying, plates were spot inoculated with each isolate and incubated for 7 days at 30 °C. Siderophore-producing bacteria form a halo around colonies due to iron chelation. Results were visually analyzed in terms of halo width against the blue medium. The experiment was performed in triplicate.

#### 2.5.5. Ammonia Production Assay

Ammonia production activity of selected PSB was performed as described by Cappuccino JC et al., (1992) [38]. Briefly, 100 μL (OD_600nm_ = 0.8) of each bacterial suspension was inoculated into tubes containing 10 mL of peptone water, incubated at 30 °C and shaken at 150 rpm for 96 h. Uninoculated medium served as the negative control. Afterwards, 1 mL aliquots were taken and centrifuged at 10,000 rpm for 10 min. Next, 0.5 mL of *Nessler’s* reagent was added to each supernatant. The ammonia production was considered positive following the development of a brownish coloration and absorbance was measured at 450 nm. The concentrations of ammonia were estimated using a standard curve of ammonium sulphate for concentrations in 0–0.3 µmol/mL range [39].

#### 2.5.6. HCN Production Assay

The PSB isolates were tested for hydrogen cyanide (HCN) production by adapting the method of Lorck et al. (1948) [40]. Briefly, TSA medium was amended with 0.44% glycine and 100 μL (OD_600nm_ = 0.6) of each strain was flooded on poured agar plates using a sterilized glass spreader. A Whatman filter paper was soaked in 0.5% picric acid in 2% sodium carbonate for 1 min and stuck below the plates’ lids. The Petri dishes were sealed with parafilm and incubated at 30 °C for 96 h. HCN production was monitored on Whatman paper following color shift from yellow (conferred by sodium picrate solution) to orange or brown. The experiment was performed in triplicate.

#### 2.5.7. Extracellular Enzymes Production

Selected PSB were qualitatively assessed for protease and cellulase production. Proteolytic activity was analyzed according to the method of Kavitha et al. (2013) [41] using a medium containing (in 1 L of distilled water): 5 g pancreatic casein, 2.5 g yeast extract, 1 g glucose and 15 g agar. The pH of the medium was adjusted to 6.75 ± 0.25 and then autoclaved. After cooling, 100 mL of a 10% sterilized skim milk solution was added to the medium which was then seeded by spot inoculation method. The clear zones around colonies appearing after 48 h indicate positive proteolytic activity. To check cellulase activity, bacteria were incubated on mineral salt agar plates containing 0.4% (NH_4_)_2_SO_4_, 0.6% NaCl, 0.1% K_2_HPO_4_, 0.01% MgSO_4_, 0.01% CaCl_2_, 0.5% carboxymethyl cellulose sodium salt (CMC) and 2% agar. At the end of the incubation, 1% Congo Red solution was poured on the surface of grown culturesduring 20 min. Next, plate surfaces were flooded with 1 M NaCl solution and left to stand for 30 min. The appearance of halos around colonies indicates the degradation of CMC and reflects cellulase production [42]. The experiments were carried out in triplicate and the diameters of the halos were measured in centimeters.

### 2.6. In Vivo Assessment of Selected PSB Strains

#### 2.6.1. Seed Germination Assay

Quinoa (*Chenopodium quinoa Willd.*) seeds were firstly sorted to eliminate those with damageable aspects. Next, seeds were surface sterilized with 2% sodium hypochlorite solution for 1 min, shaken in 70% ethanol for 1 min and washed 5 times in sterilized distilled water followed by air-drying under laminar flow hood. The bacterial pellets (OD_600nm_ = 0.8) were obtained from fresh cultures by centrifugation at 10,000 rpm for 5 min. Each bacterial pellet was resuspended in 10 mL of sterile distilled water, vortexed for 10 s and used for seed treatment. Bacterial suspensions were applied as seed drenches with a ratio of 10 mL per 90 seeds for 1 h [43]. Afterwards, seeds were air-dried and placed on plates containing 0.7% sterilized agar [44]. Thirty seeds were placed on each plate. Triplicates of each treatment were maintained and seeds treated with sterilized distilled water were used as negative control [34]. The plates were maintained at 25 °C for 48 h in the dark [43] and germination rates were monitored 24 h and 48 h post-incubation. After three days, the plates were maintained at room temperature in a day/night cycle (~12/12 h) for an additional 72 h and the total length and fresh and dry weight were measured. The germination rate and vigor index were computed according to the following equations [45]:Germination rate (%) = (Number of germinated seeds ÷ Total number of seeds) × 100(1)
Vigor index = Germination rate (%) × Total seedling length (cm)(2)

#### 2.6.2. In Vivo Experiment under Saline Conditions

An experimental pot study was conducted under shade house conditions using selected bacteria as bioinoculants. Quinoa (*Chenopodium quinoa Willd.*) seeds were inoculated with individual strains as outlined for the seed germination essay. We used plastic pots (height 18 cm, diameter 20 cm) containing 5 kg of substrate composed of mixture of sterilized sand and agricultural soil (3:1). Fifteen inoculated quinoa seeds were sown at ~2 cm depth in each pot with four replications per treatment arranged in complete randomized design (CRD). Ten days after sowing, the number of seedlings were reduced to two per pot. At the two-leaf stage, each pot was inoculated with 20 mL of corresponding bacterial suspension (OD_600nm_ = 0.8). Pots were watered once a day with tap water until the start of saline irrigation. Bacterial and irrigation treatments performed in this experiment are described in Table 1.

At the five-leaf stage, quinoa plants were watered daily with two irrigation treatments. The first set of pots was exposed to non-saline irrigation (0 Mm NaCl) with tap water, while the second was irrigated by saline solution of 400 mM NaCl [46], corresponding to 2.32% NaCl. The concentration of saline irrigation solution was progressively increased [46] in 100 mM increments reaching 400 mM NaCl, maintained until plant harvesting. To avoid drought stress, watering was done so as not to exceed the water ooze from the pots’ bottoms [47].

##### Plant Harvest and Phenotypes Monitoring

Following 45 days of growth, leaf chlorophyll content index (CCI) was recorded using a chlorophyll content meter (*Hansatech instruments, Model CL-01*). At day 46, plants were harvested, and the soil was rinsed off from the roots under tap water. Length and weight of the roots and shoots were determined as well as leaf area using *Petiole* mobile application [48]. Dry weights of samples were recorded post oven-drying at 70 °C for 48 h.

##### Plant Nutrient and Ionic Analysis

The oven-dried plants were separately ground and used as matrices for determination of P, K, Na^+^, K^+^ and Ca^2+^ contents. The concentration of each element was determined using optical emission spectrometry coupled with inductively coupled plasma (*Agilent 5110 ICP-OES*). The results are expressed as percentage of dry matter (% DM). Quadruplicate sets were performed for each treatment.

### 2.7. Statistical Analysis

All obtained data were statistically analyzed using IBM SPSS Statistics 20 software. Comparison between treatments were performed using one-way analysis of variance (ANOVA) followed by post-hoc analysis with Tukey test. The level of statistical significance was set at *p* < 0.05. 

## 3. Results

### 3.1. QA1 and QF11 Are Strains Exhibiting High Phosphate Solubilization Activities 

Bacteria isolated from the Quinoa rhizosphere were diluted and plated on TSA. Seventy-nine single colonies were screened for P solubilization following spot inoculation on NBRIP agar medium plates containing TCP as the sole source of phosphate. Based on the appearance of surrounding solubilization halos, two isolates named QA1 and QF11 were selected for further studies. To quantify P solubilization, the two strains were grown in liquid NBRIP broth supplemented with TCP for 5 days (see Material and Methods: M&M). The two QA1 and QF11 strains showed high capacities to solubilize P: 346 ± 8.71 mg/L and 220 ± 8.71 mg/L, respectively (Figure 1A). Additionally, the P solubilization efficiency of QA1 and QF11 correlated negatively with decreased pH of the culture supernatants (Figure 1A). 

### 3.2. QF11 Strain Overproduces Indole-3-Acetic Acid (IAA)

Production of IAA by bacteria is considered one of the remarkable PGPR features. QA1 and QF11 strains were assessed, in vitro, for their capacity to produce IAA in TSB medium supplemented with 0.1% L-tryptophan as a metabolic precursor for IAA synthesis (see Materials and Methods). Seven days post incubation, both tested strains were able to produce IAA with significantly various concentrations as illustrated in Figure 1B. The highest amount was obtained in QF11, 795.31 ± 80.06 µg/mL, while QA1 produced 180.5 ± 2.22 µg/mL.

### 3.3. Strain QA1 Tolerates High Salt Concentrations

Salinity is an abiotic stress that affects bacterial development. To assess the ability of QA1 and QF11 to tolerate salt stress conditions, bacteria were first grown with increasing NaCl concentrations in TSA. We found that the growth of QF11 was inhibited at 900 mM NaCl while QA1 tolerated 2000 mM NaCl final concentration (Table 2). Subsequently, salt stress tolerance was tested in liquid medium. Bacteria grown in TSB containing increasing NaCl concentrations were quantified by measuring the OD_600nm_ following 48 h of incubation. We found that QF11 showed low salt stress tolerance, with a maximum growth observed at 0 mM NaCl and decreasing bacterial mass with increasing salt concentrations, until a total growth inhibition at 1800 mM NaCl. Comparatively, the maximal growth of QA1 strain was seen at 300 mM NaCl, while its total growth arrest was detected at 2400 mM NaCl (Figure 2B). 

### 3.4. QA1 Strain Overproduces Siderophores And QF11 Enhances Biofilm Formation 

Siderophores are iron chelators of dual interest; they are an important source of assimilable iron for plants and participate in spatial colonization against phytopathogens. Thereby, we investigated the ability of our two strains to produce siderophores on the CAS-Agar medium (see Materials and Methods). We found that the amount of siderophores was remarkably higher in QA1 compared to QF11 strain (Table 2 and Figure 3D). 

Biofilm formation by bacteria is an important phenomenon that plays numerous roles in many physiological processes [49]. Here, using the crystal violet binding assay (see Materials and Methods), we demonstrated that both QA1 and QF11 strains produce high amounts of biofilm (Figure 2A). The OD_600nm_ of the negative control did not exceed 0.03. 

### 3.5. Strain QA1 Overproduces Ammonia and Hydrogen Cyanide (HCN)

Ammonia and HCN are chemical compounds exerting various benefits on plant health mainly by acting as metabolic inhibitors against phytopathogens. The two isolates were able to produce ammonia but the highest amount (0.7 μmol/mL) was detected using the QA1 strain (Figure 2A). Similarly, QA1 produced higher amounts of HCN compared to QF11 (Table 2). 

### 3.6. Strain QA1 Overproduces Extracellular Enzymes

Bacterial extracellular enzymes such as proteases and cellulases are involved in soil fertilization and the biocontrol of phytopathogens through microbial membrane degradation [50]. Therefore, we monitored protease and cellulase production by the two strains (Figure 3B,C). In the cellulase production assay, the ratio “halo diameter/colony diameter” was 4.21 for QA1. Protease was also produced by QA1 (Figure 3C) with a diameter rate of 1.07 (Table 2). Neither protease nor cellulase activities were detected using QF11 strain.

### 3.7. QA1 and QF11 Belong to the Genus of Bacillus Licheniformis and Enterobacter Asburiae, Respectively

Phylogenetic analysis of generated DNA sequences of the two strains by 16S rRNA gene sequencing using NCBI and SILVA databases revealed that QA1 correspond to *Bacillus licheniformis* (MN810040) with degrees of 99% proximity, while QF11 is closely related to *Enterobacter asburiae* (MN810041) with 99.8% similarity (Figure 4). Further *cpn60* gene sequencing allowed us to confirm QA1 strain as *B. licheniformis*. 

### 3.8. Strain QF11 Increases Seed Germination

Inoculation of seeds, soil or plants by PGPRs, such as PSBs, is a promising approach to improve global agricultural production and enhance nutrient use efficiency [51]. The bacterial treatment of quinoa seeds showed that our two strains exerted a significant positive influence on early plant growth, namely on germination rate, total length and fresh and dry weights (Table 3 and Figure 5A). However, these effects varied with the two isolates. Synoptically, coated seeds with bacterial suspensions showed a significant increase in germination rate especially after 24 h of incubation, ranging from 153% to 305% as compared to control seeds (surface sterilized and then treated with sterile distilled water). The same effect was recorded on the total length of seedlings, with a significant increase ranging from 50.5% to 211%. Likewise, germinated Quinoa seeds showed 14%–38% and 10%–75.7% increase in fresh and dry weights, respectively. The significant differences among isolated strains on seedling vigor index are graphically schematized in Figure 5B. The performances were classified in the following order: QF11> QA1 > C- (Table 3 and Figure 5B). 

### 3.9. B. licheniformis QA1 Improves Shoot Biomass while E. asburiae QF11 Enhances Root Development

Interactions of bacteria within the tripartite “plant–soil–microbes” microcosm are determinants for their functional diversity. Therefore, an in vivo test is of considerable importance as it simulates field conditions. As expected, inoculated quinoa plants showed a higher growth capacity compared to uninoculated plants which produced a lower quantity of biomass (Figure 6). Interestingly, QF11 significantly promoted root development and weight under non-saline irrigation (Figure 6 and Figure 7A,C,F) by 381.48% for root fresh weight and 61.22% for root length. Moreover, saline irrigation negatively affected plant growth compared to sets irrigated by non-saline water (Figure 7C,F).

### 3.10. B. licheniformis QA1 Strain Increases Leaf CCI, P and K+ and Decreases Na^+^ Uptake

To further characterize the effect of bacterial inoculation under salt or free salt treatments, we monitored leaf CCI. QA1 significantly affected the CCI of leaves. Under the first salt-free treatment, the QA1 was performant with an improvement of 60% in CCI. Following saline irrigation, a significant increase was recorded using QA1 (110.37%), but this difference was however not statistically significant (Figure 7H).

The level of P in plants is a critical growth parameter. Thus, we analyzed its content and revealed, that under free-saline treatment, bacterial inoculation improved P mobilization to plants. P uptake was higher, 41.17%, upon inoculation with QA1 compared to uninoculated plants. In contrast, no significant differences were observed under saline irrigation compared to control plants (Figure 8A).

To evaluate the effect of our bacteria in the mitigation of salt impact, we next studied the ionic balance in Na^+^, K^+^ and Ca^2+^ ions within inoculated quinoa plants. There was no observed difference in Na^+^ and K^+^ uptake under non-saline irrigation compared to control plants (Figure 8B,C). However, under saline irrigation, bacterial inoculation induced a significant effect on Na^+^ and K^+^ content. In fact, Na^+^ concentration decreased by 63.63% using QA1 and 28.57% using QF11, but for QF11 this difference was not statistically significant. Conversely, the K^+^ content significantly increased especially in the QA1 isolate (32.55%) (Figure 8C). As for Ca^2+^ accumulation, no significant differences were detected between the two treatments (Figure 8).

## 4. Discussion

Plants develop strategies to reduce adverse environmental effects by attracting diverse beneficial microbes in their rhizosphere [52]. Bidirectional interactions between the plants and their rhizosphere can mitigate the effect of stressful conditions [53,54,55]. A bulk of studies have suggested that the presence of PSBs can increase a plant’s tolerance to various abiotic stresses (e.g., drought, salinity, and nutrient deficiency) [56,57]. PSB have been isolated from the rhizosphere of several plants, including wheat [58], corn [59] and rice [60]. However, only few studies have reported on the PSB associated with quinoa roots [46,61]. Here, for the first time, we isolated native PSB associated with quinoa roots from Moroccan soils and assessed their role in quinoa plant growth under variable soil salinity conditions. Out of 79 isolates, two strains (QA1 and QF11) have been characterized for their potential PGBR properties and genotyping analysis revealed that QA1 and QF11 correspond to *B. licheniformis*, and *E. asburiae*, respectively.

To reduce abiotic stresses such as salinity, plants naturally trigger several physiological responses including phytohormone synthesis, antioxidant production and nutrient uptake regulation. Increased soil salinity induces Na^+^ accumulation in the plant tissues, which negatively affects plant growth by increasing the level of reactive oxygen species (ROS) that inhibit photosynthesis. Plant salinity stress is underpinned by the adverse effect of ROS on cellular protein oxidation and DNA mutation, leading to alterations of either cellular integrity, gene expression or toxic reactions [62]. Previous experimental findings indicated that PGPR could assist plants to tolerate increased salinity [4]. PGPR are known to aid the maintenance of a propitious equilibrium between Na^+^ and K^+^ ions and therefore protect plants from salt stress damage. To counter the effect of salinity on the plant, PGPR also produce protective molecules (e.g., auxins, gibberellins, cytokinins, proline and pyrroquinoline quinone (PQQ)) [63]. Increased production of these protective molecules by PGPR can result in an accentuated nutrient uptake by host plant grown in salt-exposed soils [64]. 

We have shown, in this study, that QA1 and QF11 exhibited PGPR properties. Inoculating quinoa plants with QA1 or QF11, under non-saline conditions, enhanced plant growth (Figure 7A–H). The PGPR properties of QA1 and QF11 strains are translated by their capacity to increase P availability to the plant (Figure 8A). The highest level of soluble P was detected in the culture supernatant of *B. licheniformis* QA1. Not surprisingly, P solubilization was reflected by medium acidification, most likely due to a low amount of organic acids release by bacteria [65]. 

QA1 and QF11 both produced siderophores. However, QA1 showed a higher performance in producing siderophores compared to QF11 (Figure 3). Siderophores are known to enhance P availability [9], as they solubilize minerals and chelate heavy metals, which in turn increases nutrient uptake and plant growth [66]. Whipps in 2001 reported that the low iron availability in soil would suppress plant pathogenic fungi [67]. In agriculture, iron chelation through bacterial siderophores is a desired trait of dual interest: on one hand, it represents an important source of iron for the plants; on the other hand, it represents a competitive trait for spatial colonization against phytopathogens [68]. Jetiyanon et al. (2015) have shown that the catechol siderophore enterobactin produced by *Enterobacter asburiae* strain RS83 enhances plant growth [69]. Plant growth is also promoted by nitrogen (N) derivatives, especially ammonia, as they provide a direct source of N to the plant [70]. In line with this, *Enterobacter* species have been previously reported to fix nitrogen [71,72]. In our study, we found that QA1 and QF11 both produced HCN and ammonia, which counteract a wide range of fungi-causing diseases in plants [73,74]. Pathogenic diseases also control involved bacterial production of proteases and cellulases. We showed that *B. licheniformis* QA1, but not QF11, produce both enzymes, suggesting that it counteracts plant soil salinity by transforming soil composition at both biological and chemical levels [75], likely via microbial metabolites actions [50]. 

As previously reported, PGPR inoculation of seeds, soil or plants is considered as an innovative method to enhance agricultural yield [51]. We demonstrated here that QA1 and QF11 grew on plates up to 2000 mM and 900 mM of NaCl, respectively. However, in liquid media, the maximal growth was reached at 300 mM NaCl for QA1 while the growth arrest was observed at 2400 mM NaCl. A recent report showed that optimum growth conditions for *Bacillus megaterium*, *Staphylococcus haemolyticus* and *Bacillus licheniformis* occurred at 300 mM of NaCl concentration, while 1500 mM NaCl led to growth arrest [34]. Comparatively, it appears clearly that QA1 strain looks more tolerant to salinity stress.

Given that QA1 and QF11 are halotolerant and exhibit PGPR traits, we investigated their role on quinoa plant growth under saline conditions. Indeed, PGPR could be a considerable boost to plant growth under harsh conditions (e.g., soil salinity) via the release of growth-promoting substances and regulators [18]. We have not observed a significant difference in plant growth and yield parameters under 400 mM NaCl treatment, except on the root system, P assimilation and Na^+^ and K^+^ absorption. (Figure 7B and Figure 8). Salt accumulation in the rhizosphere induces negative water potential in the root system by reducing water absorption [76]. In accordance with this, significantly lowered root fresh and dry weights (Figure 7C,F), were seen in quinoa plants. QA1-inoculated plants displayed higher CCI of leaves, suggesting a stronger photosynthetic capacity. The maintenance of growth efficiency of inoculated plants strongly suggests that, despite the saline treatment, the photosynthetic apparatus is not substantially damaged [77]. Salt stress is generally reflected by an increased ionic toxicity, which in turn induces protein conformational changes owing to K^+^ replacement by Na^+^. Several enzymes require high concentration of K^+^ as a cofactor. In addition, K^+^ is also involved in protein synthesis by promoting tRNA interaction to ribosome [78]. We highlighted the increased levels of K^+^ in the DM under saline stress compared to non-saline treatment (Figure 8C) and such a finding is consistent with previous studies on quinoa and wheat [79,80,81]. Another study has suggested that quinoa plants use Na^+^ for osmotic adjustment to maintain a favorable K^+^/Na^+^ ratio content in the plant [82]. It is also known that halotolerant bacteria can accumulate osmolytes under stress conditions. Indeed, Qurashi and Sabri (2013) [83] reported that the halotolerant bacterial strains, *S. haemolyticus* and *B. subtilis*, isolated from saline rhizosphere of chickpea can moderately accumulate endogenous osmolytes, such as betaine, glycine and proline, and enhance plant growth by reducing salt stress. 

Salt stress also causes hormonal dysregulation. Our two strains have the capacity to produce IAA (e.g., strains QA1 and QF11 produced 795.31 μg/mL and 180.5 μg/mL of IAA, respectively) leading to root elongation. Zhao et al. (2011) [84] have shown that *Enterobacter asburiae HPP16* strain produced 170 mg/L of IAA in culture suspension post 35 h incubation [69]. The level of IAA produced by QA1 is also consistent with reported data in *Bacillus licheniformis ML3* (e.g., 174.72 µg/mL) [85]. IAA, being the first class of phytohormones to be identified, virtually regulate all physiological aspects of plant development and are used as a marker to select beneficial bacteria [86]. Furthermore, auxin production is known to stimulate root development, which results in better absorption of water and nutrients from the soil [87]. Interestingly, QF11 significantly promotes quinoa root development and weight under non-saline irrigation by 381.48%, 79.64% and 61.22% for root fresh weight, dry weight and root length, respectively. The remarkable effect of QF11 on root development is likely attributed to the increased IAA production (Figure 6 and Figure 7A,C,F). Comparatively, our results support the role of IAA in inducing morphological changes, increased length of the root as well as number of root hairs/laterals, which are crucial for nutrient uptake [88]. In addition, *Bacillus*, *Pseudomonas* and *Enterobacter* applied as bioinoculants promote plant growth and biomass through production of phytohormones (e.g., auxins and cytokinins) [66,89]. In this aspect, growth and grain yield improvement of wheat plants were associated with PSB inoculation and phytohormone-producing strains such as *Azospirillum*, *Bacillus* and *Enterobacter* [90].

In the present study, reducing the effect of Na+ accumulation in quinoa plants post inoculation with QA1 and QF11 could be also attributed to Na^+^ chelation by exopolysaccharides (EPS), which therefore restrict uptake by the plants [91]. Consequently, the reduced Na^+^ availability increased K^+^ absorption and improved water acquisition. In the rhizosphere, PGPR are often organized into biofilms [49]. Accordingly, we showed that QA1 and QF11 form biofilms that are mainly composed of EPS, suggesting that this may engender plant resistance to salinity [92]. Such a body of evidence clearly suggests that QA1 is the most effective strain in promoting plant biomass, reducing Na^+^ accumulation and enhancing K^+^ uptake. Comparatively, our data are in-line with previous findings [93] whereby wheat inoculation with *Bacillus licheniformis HSW-16* showed a significant tolerance reaching up to 200 mM NaCl.

## 5. Conclusions

To the best of our knowledge, the present study reports the first isolation, identification and characterization of PSB strains from the rhizosphere of quinoa isolated from Moroccan soil. PSB strains were tested for their multiple PGPR activities and were identified as *B. licheniformis* and *E. asburiae*. The two identified halotolerant strains showed promising results in terms of phyto-benefic properties. Treated quinoa seeds with the two strains (*B. licheniformis* and *E. asburiae*) did significantly improve germination rate and seedling height and weight. *E. asburiae* QF11 was found here to be the most efficient strain for promoting early plant growth. QA1 also improved quinoa growth and reduced Na^+^ uptake. Nevertheless, more in-depth studies would elucidate the impact of bacterial inoculation on the phytohormonal profile, antioxidant activity and soil properties. Further investigations would shed light on the role of *B. licheniformis* and *E. asburiae* in halotolerance. Bacterial transcriptional analysis might lead to the identification of regulated genes associated with growth-promoting effects. Translating our laboratory findings on QA1 and QF11 using not-sterilized soils on pots and into field experiments upon quinoa plant inoculation is paramount. Finally, our findings strongly suggest the use of *B. licheniformis* QA1 and *E. asburiae* QF11 strains as suitable biofertilizers.

## Figures and Tables

**Figure 1 microorganisms-08-00948-f001:**
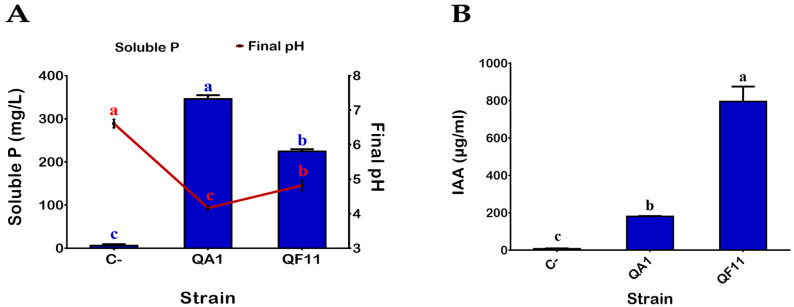
(**A**) Phosphate solubilization by selected PSB (phosphate solubilizing bacteria) in the presence of 0.5% hydroxyapatite in NBRIP broth. Drop in pH is indicated by the red line. (**B**) IAA (indole-3-acetic acid) production by selected PSB in the presence of 0.1% L-tryptophan. Uninoculated media were used as a negative control (C-). The values represent means of replicates (*n* = 3) ± standard deviations. The different letters in superscript (a, b, c…) indicate the statistically significant difference at 95% between treatments.

**Figure 2 microorganisms-08-00948-f002:**
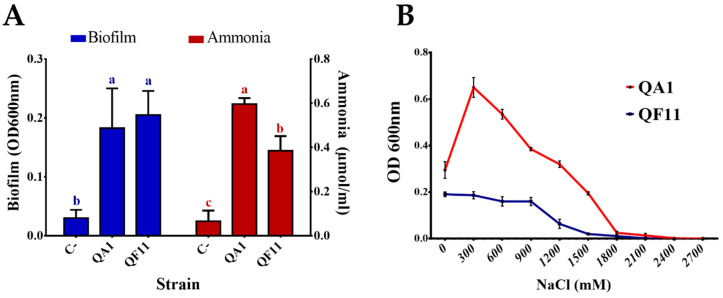
(**A**) Biofilm and ammonia production by selected PSB isolates. (**B**) Effect of NaCl on selected PSB strains after 48 h of incubation in TSB broth. The values represent means of replicates (*n* = 3) ± standard deviations. Uninoculated media were used as a negative control (C-). The different letters in superscript (a, b, c…) indicate the statistically significant difference at 95% between treatments.

**Figure 3 microorganisms-08-00948-f003:**
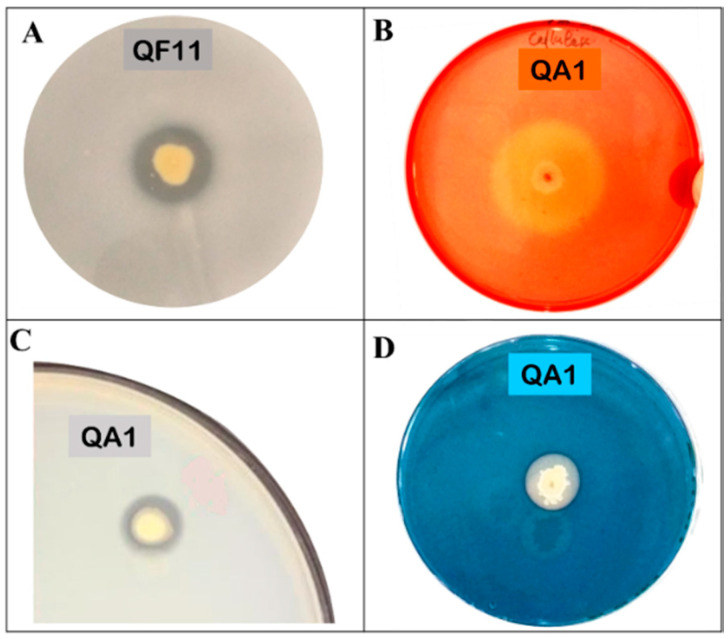
Illustration of tests performed on plates for the assessment of PGP (plant growth promoting) traits of selected PSB. (**A**) Phosphate solubilization, (**B**) cellulase production, (**C**) protease production, (**D**) siderophores production.

**Figure 4 microorganisms-08-00948-f004:**
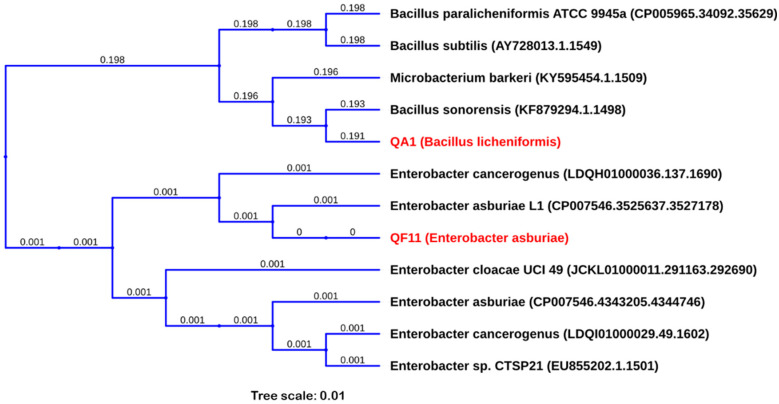
Phylogenetic tree of selected P-solubilizing bacteria, (QA1) *Bacillus licheniformis* (MN810040) and (QF11) *Enterobacter asburiae* (MN810041), based on PHYLIP Neighbor-Joining method of the 16S rRNA gene sequences using UGENE. The 16S rRNA gene sequences and tree of related species were downloaded from SILVA database.

**Figure 5 microorganisms-08-00948-f005:**
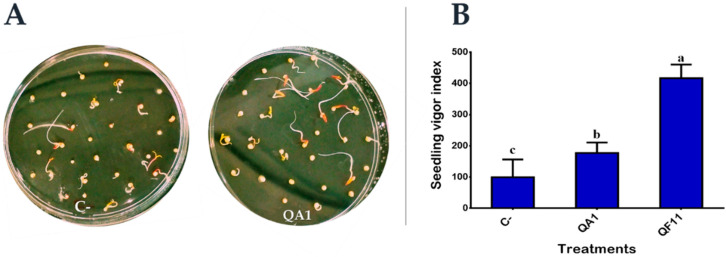
Effect of bacterial inoculation on early quinoa seeds germination and seedlings. (**A**) Germination rates monitored after 48 h of incubation in the dark at 25 °C. (**B**) Seedling vigor index of germinated quinoa seeds. C-, seeds treated with sterile distilled water; QA1, seeds treated with PSB QA1; QF11, seeds treated with PSB QF11. The values represent means of replicates (*n* = 3) ± standard deviations. The different letters in superscript (a, b, c…) indicate the statistically significant difference at 95% between treatments.

**Figure 6 microorganisms-08-00948-f006:**
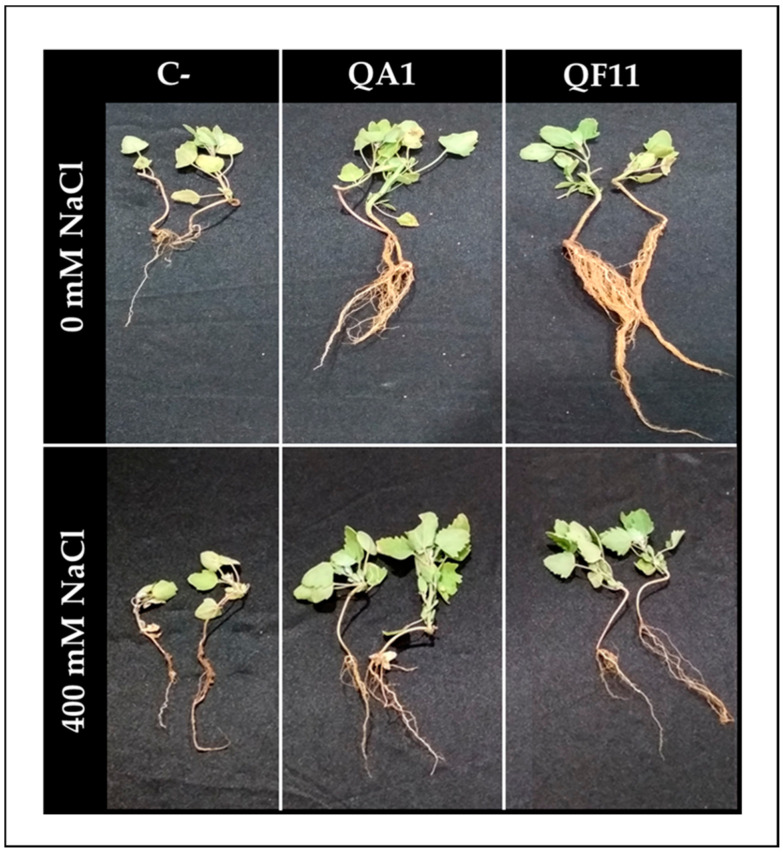
Effect of tested PSB on quinoa plant growth after 45 days of culture under normal (0 mM NaCl, tap water) and saline (400 mM NaCl) irrigations.

**Figure 7 microorganisms-08-00948-f007:**
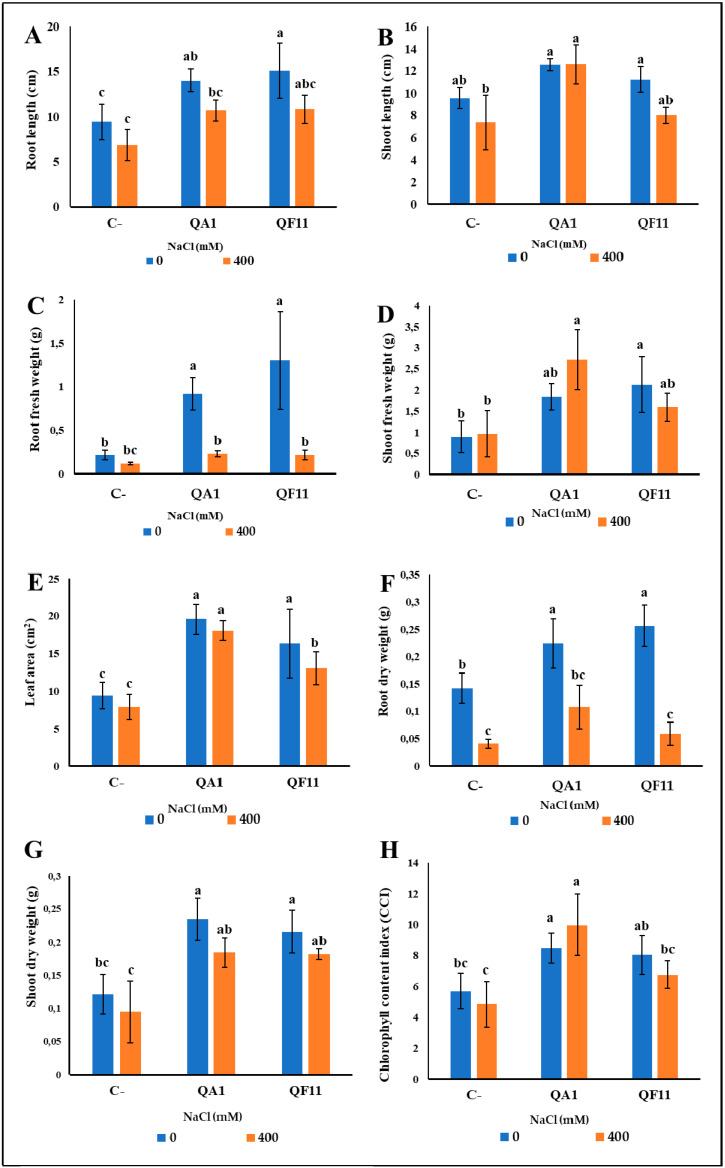
Effect of selected PSB on quinoa plant growth after 45 days of cultivation under non-saline (0 mM NaCl) and saline (400 mM NaCl) irrigations. (**A**) Root length, (**B**) shoot length, (**C**) root fresh weight, (**D**) shoot fresh weight, (**E**) leaf area, (**F**) root dry weight, (**G**) shoot dry weight, (**H**) leaf chlorophyll content index. The values represent means of replicates (*n* = 4) ± standard deviations. The different letters in superscript (a, b, c…) indicate the statistically significant difference at 95% between treatments.

**Figure 8 microorganisms-08-00948-f008:**
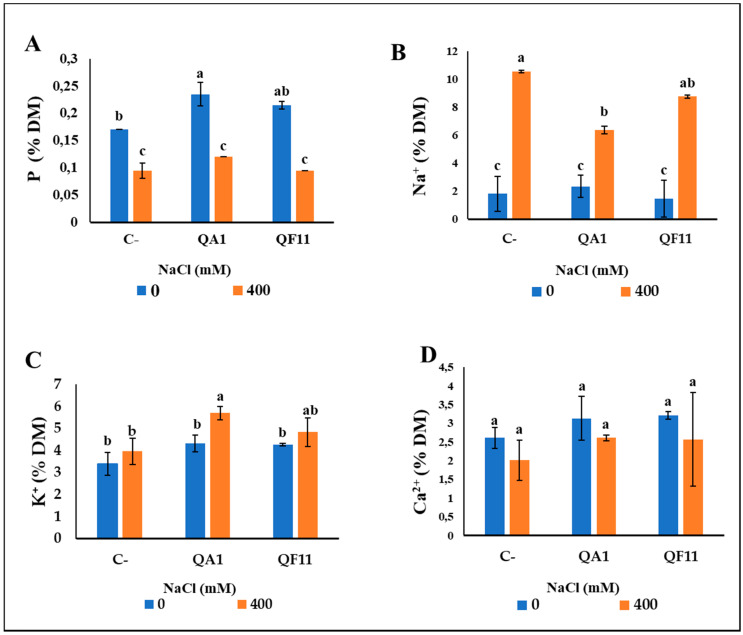
Effect of salt stress induced by NaCl and bacterial inoculation on the uptake of phosphorus (P) and salt ions by quinoa plants. (**A**) P (**B**) Na^+^ (**C**) K^+^ (**D**) Ca^2+^. The values represent means of replicates (*n* = 4) ± standard deviations. The different letters in superscript (a, b, c…) indicate the statistically significant difference at 95% between treatments.

**Table 1 microorganisms-08-00948-t001:** Pot study work plan.

0 mM NaCl		400 mM NaCl	
Symbol	Treatment	Symbol	Treatment
C-	Seeds and plants treated with sterilized distilled water (Negative control)	C-	Seeds and plants treated with sterilized distilled water (Negative control)
QA1	Seeds and plants treated with QA1 strain	QA1	Seeds and plants treated with QA1 strain
QF11	Seeds and plants treated with QF11 strain	QF11	Seeds and plants treated with QF11 strain

**Table 2 microorganisms-08-00948-t002:** Summary of relevant phenotypic traits observed with selected PSB strains using plate assay.

Bacterial Strain	Bacterial Treatment	QA1	QF11
Extreme properties	NaCl tolerance on plates	2 M	<0.9 M
	Temperature tolerance	55 °C	37 °C
Siderophore production		+++	++
HCN production		+++	+
Extracellular enzymes (*Halo Colony diameter*)	Protease Cellulase	1.07 ± 0.08 4.21 ± 0.24	− −

The ‘+’ and ‘−’ signs indicate efficiencies as follow: −, negative result; +, weakly positive; ++, moderately positive; +++, highly positive.

**Table 3 microorganisms-08-00948-t003:** Effect of strains QA1 and QF11 on quinoa seed germination parameters.

Parameter	Incubation Time	C-	QF11	QA1
Germination rate (%)	24 h	16.6 ± 6.65 ^c^	67.3 ± 12.5 ^a^	43.2 ± 6.7 ^b^
	48 h	58.6 ± 5.13 ^c^	77.6 ± 6.8 ^a^	67.7 ± 5.09 ^b^
Total length (cm)		1.7 ± 0.55 ^c^	5.3 ± 0.32 ^a^	2.56 ± 0.94 ^bc^
Fresh weight (mg)		36 ± 10.5 ^c^	50 ± 10.8 ^a^	41.3 ± 3.51 ^bc^
Dry weight (mg)		6.6 ± 1.52 ^b^	11.6 ± 2.51 ^a^	7.3 ± 1.52 ^ab^

C- (Negative control), seeds treated with sterile distilled water; QA1, seeds treated with PSB QA1; QF11, seeds treated with PSB QF11. The numerical values represent means of replicates (*n* = 3) ± standard deviations. The different letters in superscript (a, b, c…) indicate the statistically significant difference at 95% between treatments.

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
