# Peer review of "Plant Growth Enhancement using Rhizospheric Halotolerant Phosphate Solubilizing Bacterium *Bacillus licheniformis QA1* and *Enterobacter asburiae QF11* Isolated from *Chenopodium quinoa* Willd"

_microorganisms, 2020, doi:10.3390/microorganisms8060948_

Round 1
Reviewer 1 Report
In this revised version, the authors have taken into account all my minor suggestions. Thus, I think the manuscript is ready for publication !
Reviewer 2 Report
I don't find any significant change in the resubmitted version, however, in the round 2 of revision I stated that the authorsdealt with all required changes and responded to all reviewer's comments.So, I accept the manuscript in the present form.
This manuscript is a resubmission of an earlier submission. The following is a list of the peer review reports and author responses from that submission.
Round 1
Reviewer 1 Report
Comments
The manuscript by MAHDI and collaborators reports the screening ofphosphate solubilizing bacteria isolated from the rhizospheric soil of Chenopodium Quinoa Willd and assessed for their PGPR properties including production of indole‐3‐acetic acid (IAA), siderophores, hydrogen cyanide (HCN), ammonia, and extracellular enzymes.We also investigated their tolerance to salt stress, and their capacity to form biofilms. From these measurements, the authors identified two bacterial strains, named QA1 and QF11, displaying a high efficiency to release soluble P from insoluble calcium apatite and a high tolerance to NaCl, among other properties revealed in vitro. In addition these two strains inoculated to Quinoa seeds were able to induce higher germination rate and increased seedlings early growth, and improve their salt tolerance. These two bacterial strains belong to the genus Bacillus licheniformis (QA1) and Enterobacter asburiae(QF11). Taken together, the study is well designed and the results are well presented and discussed.
I have only minor editorial comments that should be done before to be accepted for publication.
Minor comments
Line 87: add “to” before mitigate
Line 87-88: write “Quinoa (Chenopodium Quinoa Willd.), an herbaceous plant species belonging to the Amaranthaceae family, is renowned ….” Instead of “Quinoa (Chenopodium Quinoa Willd.) herbaceous plant species belonging to the Amaranthaceae family is renowned ….”
Line 130: should we read “colorimetrically” instead of “calorimetrically”
Line 167: please check the value “02 μL of each bacterial”
Lines 259, 291, 293, 299, 303, 317, 327, 332, 348, 350, 360, 366, 378, 386, 387, 404, 406, 408, 410, 426, 436, 439, 440, 464, 474: please give the adequate citation to replace “Error! Reference source not found” given repeatedly throughout the manuscript.
Line 354 and 358: delete the portion of the figure legend inserted in the main text.
Line 440: delete “is” at the end of the sentence.
Author Response
Reviewer 1: The manuscript by MAHDI and collaborators reports the screening of phosphate solubilizing bacteria isolated from the rhizospheric soil of Chenopodium Quinoa Willd and assessed for their PGPR properties including production of indole‐3‐acetic acid (IAA), siderophores, hydrogen cyanide (HCN), ammonia, and extracellular enzymes. We also investigated their tolerance to salt stress, and their capacity to form biofilms. From these measurements, the authors identified two bacterial strains, named QA1 and QF11, displaying a high efficiency to release soluble P from insoluble calcium apatite and a high tolerance to NaCl, among other properties revealed in vitro. In addition, these two strains inoculated to Quinoa seeds were able to induce higher germination rate and increased seedlings early growth and improve their salt tolerance. These two bacterial strains belong to the genus Bacillus licheniformis (QA1) and Enterobacter asburiae(QF11). Taken together, the study is well designed, and the results are well presented and discussed.
We thank the reviewer for his positive comment.
I have only minor editorial comments that should be done before to be accepted for publication.
Minor comments
Q1: Line 87: add “to” before mitigate
A1: Done in line 86
Q2: Line 87-88: write “Quinoa (Chenopodium Quinoa Willd.), an herbaceous plant species belonging to the Amaranthaceae family, is renowned ….” Instead of “Quinoa (Chenopodium Quinoa Willd.) herbaceous plant species belonging to the Amaranthaceae family is renowned ….”
A2: Done: line 86
Q3: Line 130: should we read “colorimetrically” instead of “calorimetrically”
A3: Done: line 130
Q4: Line 167: please check the value “02 μL of each bacterial”
A4: This value is correct, for clarification 02 is now replaced by 2, text line 167.
Q5: Lines 259, 291, 293, 299, 303, 317, 327, 332, 348, 350, 360, 366, 378, 386, 387, 404, 406, 408, 410, 426, 436, 439, 440, 464, 474: please give the adequate citation to replace “Error! Reference source not found” given repeatedly throughout the manuscript.
A5: We apologize for these errors, corrected all along the text.
Q6: Line 354 and 358: delete the portion of the figure legend inserted in the main text.
A6: Done.
Q7: Line 440: delete “is” at the end of the sentence.
A7: Done
Reviewer 2 Report
This work deals with two aspects:
-The selection and characterization of the two bacteria solubilizing Phosphorus named QA1 and QF11.
-The role of these two bacteria on the tolerance of Quinoa to salinity.
Generally, this manuscript is well written, the problem is well mentioned, however, a hypothesis remains to be defined.
My major remarks:
- This manuscript treated several parameters, however, the results and the discussions were presented in a descriptive way, I encourage the authors to make an integrative discussion of all the results.
-In the discussion section, please compare the values of solubilized phosphate up to 346 mg / L and the production of IAA up to 795.31μg / mL with previous results in the literature.
- Why did the authors choose sterilized soil in their experiment on in vivo experiment under saline conditions?
Minor remarks
References are missing in the text
Author Response
Reviewer 2
This work deals with two aspects:
-The selection and characterization of the two bacteria solubilizing Phosphorus named QA1 and QF11.
-The role of these two bacteria on the tolerance of Quinoa to salinity.
Q1: Generally, this manuscript is well written, the problem is well mentioned, however, a hypothesis remains to be defined.
A1: The working hypothesis is now clarified in the last paragraph of the introduction section.
Major remarks:
Q2: - This manuscript treated several parameters; however, the results and the discussions were presented in a descriptive way, I encourage the authors to make an integrative discussion of all the results.
A2: We thank the referee for this remark. The discussion is now reorganized and presented as in integrative way.
Q3: In the discussion section, please compare the values of solubilized phosphate up to 346 mg / L and the production of IAA up to 795.31μg / mL with previous results in the literature.
A3: The comparison of IAA production is now added in the discussion section (lines515 - 516). We removed the values of P solubilization in the discussion section because they are close to those described in literature.
Q4: Why did the authors choose sterilized soil in their experiment on in vivo experiment under saline conditions?
A4: This remark also pointed by reviewer to investigate the single implication of QA1 and QF11 inoculum. We agree that, for a large application on the field, our inoculum will interact with other rhizospheric bacteria. We are planning to conduct other studies using natural no sterilized soil both on pots and on the field. We expect a synergetic effect with enriched QA1 and QF11 inoculum.
Minor remarks
Q5: References are missing in the text
A5: We apologize, they are corrected now
Reviewer 3 Report
The authors studied the Halotolerant Phosphate Solubilizing Bacterium Bacillus licheniformis Isolated from Chenopodium quinoa. The topic is highly relevant from both scientific and practical points of view. However, The question behind the work is not clear and the title is too long and confusing. The manuscript deals with characterizing the bacterium Bacillus licheniformis rather than its effect on enhancing salinity tolerance in quinoa plants. Moreover, the authors claimed that “Reports on halotolerant PSB colonizing Quinoa plant rhizosphere and their effects on plant growth have not yet been reported.” It is not true. The bacteria associated with the cultivation of quinoa were investigated in Ortuño et al. 2014 in Revista de Agricultura, Enhancing the Sustainability of Quinoa Production by phosphate solubilizing bacteria was studied by Ortuño et al., 2013 in Agronomy, and Enhancing salt tolerance in quinoa by halotolerant bacterial inoculation was studied earlier in Yang et al. 2015 in Functional Plant Biology (the reference you cite as [46]).
I have gone through the manuscript several times and tried to find some novelty and addition of knowledge to cited aspect. Poor language and style of writing are so varied ranging from typos to odd word choices or concocted words, syntax and grammatical errors, perverted phrases and narratives, so it was a daunting task to reach the end. The manuscript is also very haphazard, with repeated sentences. For example: “Salinity is a limiting factor for crop production in semi‐arid …”, “Salinity is a major factor in reducing crops production”, “Indeed, arid and salt exposed soils are known to be inadequate for agricultural cultivation”
The figures are not good and not clear enough to read, references are incomplete as there is an error in citing references at many places.
Therefore, I reject this manuscript as it is not suitable for publication in the present form.
Author Response
Reviewer 3
Q1: The authors studied the Halotolerant Phosphate Solubilizing Bacterium Bacillus licheniformis Isolated from Chenopodium quinoa. The topic is highly relevant from both scientific and practical points of view. However, the question behind the work is not clear and the title is too long and confusing.
A1: We thank the reviewer for his statement about the relevance of the topic. We clarified the working hypothesis in the last paragraph of the introduction section. As requested, we also shortened the title: Reducing Plant Salinity-induced Damage by Halotolerant Phosphate Solubilizing Bacterium Bacillus licheniformis QA1 and Enterobacter asburiae QF11 Isolated from Chenopodium quinoa Willd Rhizosphere.
Besides we are little surprised as the reviewer focused his interest on Bacillus licheniformis, while our report studied also Enterobacter asburiae QF11.
Q2: The manuscript deals with characterizing the bacterium Bacillus licheniformis rather than its effect on enhancing salinity tolerance in quinoa plants. Moreover, the authors claimed that “Reports on halotolerant PSB colonizing Quinoa plant rhizosphere and their effects on plant growth have not yet been reported.” It is not true. The bacteria associated with the cultivation of quinoa were investigated in Ortuño et al. 2014 in Revista de Agricultura, Enhancing the Sustainability of Quinoa Production by phosphate solubilizing bacteria was studied by Ortuño et al., 2013 in Agronomy, Enhancing the Sustainability of Quinoa Production and Soil Resilience by Using Bioproducts Made with Native Microorganisms. Noel Ortuño et al., 2013 and Bacteria associated with the cultivation of quinoa in the Bolivian Altiplano and their biotechnological potential Noel Ortuño., 2014
A2: We partly agree with the reviewer comments. Indeed, there is some previous studies on PSB isolated from Quinoa. The statement that report on halotolerant……..have not been reported yet is changed overall the manuscript.
The reviewers cited three references that we compared and discussed hereafter.
I/ The first one by Ortuño et al., 2013 in Agronomy, Enhancing the Sustainability of Quinoa Production and Soil Resilience by Using Bioproducts Made with Native Microorganisms
And
II/ Bacteria associated with the cultivation of quinoa in the Bolivian Altiplano and their biotechnological potential Noel Ortuño., 2014
-The first difference compared to our study, in that the authors isolated Endophytic Bacillus licheniformis from Quinoa leaves while in our study QA1 was isolated from the rhizosphere of Quinoa and was not endophytic.
- All their isolated bacteria were monitored for their PGPR traits, however their B. licheniformis strain was neither used for plant inoculation under greenhouse conditions nor in the germination assay. Rather, the authors focused their study only on bacterial isolates showing positive results. Which obviously did not include B. licheniformis. Here is an extract of the cited paper “for the three tests performed under in vitro conditions were used to inoculate plants under greenhouse conditions in order to confirm their beneficial properties in an in vivo assessment. The bacterial isolates that increased plant height were 1Bp and 5Bp (B. pumilus, P-solubilization bacteria), 139 and 143 (B. simplex, N-fixing bacteria), 2p, 3p and 4p (Paenibacillus sp., N-fixing bacteria) and BAQ-11 (B. subtilis, N-fixing bacteria) and for panicle weight were 3p and 4p (Paenibacillus sp.) and BAQ-11 (B. subtilis) (, for more details please refer to Figure 1 of paper by Ortuño et al., 2013.”
- Enterobacter asburiae, the equivalent of our QF11 strain was not at all among the isolated bacteria in both cited articles.
Thus, we conclude here that, compared to our study, the cited reports are different.
II/ Enhancing salt tolerance in quinoa by halotolerant bacterial inoculation was studied earlier in Yang et al. 2015 in Functional Plant Biology (the reference you cite as [46]).
Again, we don’t agree with the reviewer comments. This study aimed to improve the productivity of quinoa using six plant growth-promoting bacterial strains (endophytic and rhizospheric ones). Using osmoadaptation assay, the authors screened two most halotolerant strains (Enterobacter sp.(MN17) and Bacillus sp. (MN54).
MN17 was isolated from maize but not from Quinoa which contrasts with our study. Indeed, this strain was first isolated by Muhammad Naveed, 2013: “The endophyte Enterobacter sp. FD17: a maize growth enhancer selected based on rigorous testing of plant beneficial traits and colonization characteristics. It’s interesting to notice that genotype of both MN17 (Enterobacter sp) and MN54 Bacillus sp) were not precisely determined. The term Sp (species) is broad and must include different derivative genotypes raising from one specie). In our study, using two complementary approaches: 16sRNA and cpn60 genes sequencing, we identified QF11 as Enterbacter asburie and QA1 as Bacillus licheniformis.
Additional differences are listed below:
1.The duration of the study was different, in our study the plants were analyzed after 45 days while in Yang study the plants were grown until harvest,
2.We have analyzed the P, K+, Na+ et Ca2+ plant composition while Yang et al., have measured other physiological parameters.
3.Enterobacter sp.(MN17) and Bacillus sp. (MN54) are endophytic strains while QA1 and QF11 are rhizospheric strains.
Regarding these observations, it obviously appears that our study, again, is completely different from the Yang’s one.
In few lines, we would like to highlight the originality of our works
- We strongly believe that our study is a comprehensive one as we went from sampling to strains selection, then from evaluating PGPR traits (7 assays in vitro), then to study abiotic resistance (heat and salt stress) in vitro, then we switch in vivo by performing seed germination tests, and lastly to the pot experiment using sterilized soil. Our strains were isolated from the rhizosphere of quinoa itself, which is not the case in the cited articles. Furthermore, our two strains are not at all the same species as those reported in the previous three articles.
- Finally, the following additional facts support, from our point of view, the originality of our work:
- Our two species ( Licheniformis and E. asburiae) have not been previously isolated from the quinoa rhizosphere and tested in vivo on quinoa itself,
- The characteristics of the soil sampled, and the substrate used in the pot experiment are different from those used before,
- In our study, we performed numerous in vitro tests to characterize our strains for their PGP properties, including biofilm, the production of extracellular enzymes and the germination test of quinoa seeds, that are missing in the three cited references.
- The variety of quinoa and the period of the test are not the same as those reported elsewhere (45 days in our study, at least 120 days)
- To the best of our knowledge our study is the first in Morocco that addressed PGP potential halotolerant bacterial species in quinoa salt affected soils.
Q3: I have gone through the manuscript several times and tried to find some novelty and addition of knowledge to cited aspect. Poor language and style of writing are so varied ranging from typos to odd word choices or concocted words, syntax and grammatical errors, perverted phrases and narratives, so it was a daunting task to reach the end. The manuscript is also very haphazard, with repeated sentences. For example: “Salinity is a limiting factor for crop production in semi‐arid …”, “Salinity is a major factor in reducing crops production”, “Indeed, arid and salt exposed soils are known to be inadequate for agricultural cultivation”
A3: We agree with this remark. The two following sentence lines 44 and 45 have been deleted: ‘’Salinity is a major factor in reducing crops production. Indeed, arid and salt exposed soils are known to be inadequate for agricultural cultivation’’
Q4: The figures are not good and not clear enough to read, references are incomplete as there is an error in citing references at many places.
A4: The figures have been improved for their contents, resolution and clarity. The missing references are now included, it is independent of our will.
Q5: Therefore, I reject this manuscript as it is not suitable for publication in the present form.
A5: Following our responses point by point to all raised queries, we hope that the reviewer should review his negative decision.
Reviewer 4 Report
The manuscript entitled "Reducing Salinity-induced Damage of Quinoa Plant by Halotolerant Phosphate Solubilizing Bacterium Bacillus licheniformis QA1 and Enterobacter asburiae QF11 Isolated from Chenopodium quinoa Willd Rhizosphere", presents comprehensive information about the newly identified PGPR from rhizospheric soil of Quinoa and that showed tolerance to high salinity stress. The manuscript is well written and easy to understand.
However, authors may consider adding the high-resolution images for Figure 6.
Please check the statistical analysis of the Figures 7C and 7G. It is a little bit doubtful.
Still, the PGPR application is limited due to the complex interactions of the PGPR with native soil microbiota. To use this in practical application authors may consider conducting the experiment with the non-sterilized soil where the PGPRs need to compete with the native soil microbiota to show the activity.
Author Response
Reviewer 4
Comments and Suggestions for Authors
The manuscript entitled "Reducing Salinity-induced Damage of Quinoa Plant by Halotolerant Phosphate Solubilizing Bacterium Bacillus licheniformis QA1 and Enterobacter asburiae QF11Isolated from Chenopodium quinoa Willd Rhizosphere", presents comprehensive information about the newly identified PGPR from rhizospheric soil of Quinoa and that showed tolerance to high salinity stress. The manuscript is well written and easy to understand.
Q1: However, authors may consider adding the high-resolution images for Figure 6.
A1: Done, a high-resolution image of Figure 6 is now provided.
Q2: Please check the statistical analysis of the Figures 7C and 7G. It is a little bit doubtful.
A2: As requested, we performed new statistical analysis, we confirm our data presented now in new figures 7 and 8.
Q3: Still, the PGPR application is limited due to the complex interactions of the PGPR with native soil microbiota. To use this in practical application authors may consider conducting the experiment with the non-sterilized soil where the PGPRs need to compete with the native soil microbiota to show the activity.
A3: Thanks to the referee for his constructive remark, we agree with him. As pointed also by reviewer 2, we plan to reconduct this study in field on Quinoa plants to explore B. licheniformis QA1 and E. asburiae QF11 effect using saline soil from different regions in Morocco. Added now in the last paragraph of the conclusion section
Reviewer 5 Report
In this manuscript, two halotolerant bacteria isolated from quinoa rhizosphere were characterized for their PGPB properties. Both QA1 and QF11 strains were shown to be beneficial for quinoa growth.
PGPB properties were well-characterized in the manuscript, however, mechanisms of plant growth promotion by these bacteria were not clear. To link these, quinoa plants must be cultivated under specific conditions to clearly see how these bacteria help quinoa growth.
There were no significant differences in shoot dry weight (Fig.7 D, G) and leaf are (Fig. 7E) between 0 mM and 400 mM NaCl conditions without bacterial inoculation. Salinity stress conditions used seem to be mild because growth of aerial parts was not affected, therefore, higher NaCl must be used to evaluate growth parameters of quinoa under salinity stress.
Picture quality of Fig 6 is very poor and this should be replaced.
There are so many errors found throughout the manuscript and these must be carefully corrected. Followings are just some of examples:
L79: “Ca2+” and “CO32-”
L114: "Ca3(PO4)2"
L259 and elsewhere: Delete “Error! Reference source not found”
L263 and elsewhere: Use “.” for decimal point but not “,”
Author Response
Reviewer 5
Comments and Suggestions for Authors
In this manuscript, two halotolerant bacteria isolated from quinoa rhizosphere were characterized for their PGPB properties. Both QA1 and QF11 strains were shown to be beneficial for quinoa growth.
Q1: PGPB properties were well-characterized in the manuscript, however, mechanisms of plant growth promotion by these bacteria were not clear. To link these, quinoa plants must be cultivated under specific conditions to clearly see how these bacteria help quinoa growth.
A1: We totally agree with this referee’s remark. We plan to perform additional experiments under controlled conditions with nutrient solution and for a long period. We are also planning to repeat our experiments in no sterilized soil both on pots and then in the field. We are currently investigating bacterial genes transcriptomic to identity genes that are highly expressed under in salinity treatments. We strongly believe that multiples approaches will be needed to decipher how QF11 and QA1 help Quinoa growth.
Q2: There were no significant differences in shoot dry weight (Fig.7 D, G) and leaf are (Fig. 7E) between 0 mM and 400 mM NaCl conditions without bacterial inoculation. Salinity stress conditions used seem to be mild because growth of aerial parts was not affected, therefore, higher NaCl must be used to evaluate growth parameters of quinoa under salinity stress.
A2: Here, we partly agree with the statement of the reviewer that there has been no effect of salt stress or that the effect is "mild". As a proof, we can see differences at the root level on all parameters (especially those reported in Figure 7 C and F). This effect is indisputably more pronounced on the roots than on the aerial parts (even if there is a decrease which remains not statistically significant). It is very common to have different effects from salt between the roots than on the aerial parts. Importantly, Quinoa tolerates abiotic stresses such as salinity (Hariadi et al. 2011; Adolf et al. 2012,2013; Shabalaetal.2012) and drought (Garciaetal.2003; Jacobsen et al. 2009; Sun et al. 2014; Razzaghi et al. 2015). Regarding our data, the effect of stress is more pronounced on roots than on aerial parts. We cannot exclude that quinoa sets up mechanisms to reduce the effect on the aerial part (exclusion of Na + at the roots, etc.). Comparatively, Adolf et al 2013: Salt tolerance mechanisms in quinoa (Chenopodium quinoa Willd.), Environmental and Experimental Botany, Volume 92, Pages 43-54) and the references cited in this paper as (Jacobsen et al., 2001, Jacobsen et al., 2003, Koyro and Eisa, 2008, Hariadi et al., 2011). We can go up to 400 mM NaCl without problem. We believe that our two studied strains boost the plant (even without salt), resulting in better resistance to salinity.
Q3: Picture quality of Fig 6 is very poor and this should be replaced.
A3: Done, a high-resolution image of Figure 6 replaces the old one.
Q4: There are so many errors found throughout the manuscript and these must be carefully corrected. Followings are just some of examples:
L79: “Ca2+” and “CO32-”, L114: "Ca3(PO4)2", L259 and elsewhere: Delete “Error! Reference source not found”
L263 and elsewhere: Use “.” for decimal point but not “,”
A4: We apologize for the errors. Corrected all along the text.
Round 2
Reviewer 2 Report
In this new version of the manuscript, the authors improved the quality and interpretation of the results, and all my comments were taken into consideration.
Author Response
We thank the reviewer for his positive conclusion.
Reviewer 3 Report
The authors dealt with all required changes and responded to all reviewer's comments.
Now the manuscript is suitable for publication.
Author Response
We thank the reviewer for his positive conclusion,
Reviewer 5 Report
This is the revised manuscript describing characterization of PGPRs and their effects on quinoa growth. The authors have revised the manuscript by following the reviewer’s comments, however, there is still one major concern which must be carefully revised.
The response to my comments #2 is misleading. The title is “Reducing Plant Salinity Induced Damage…”, however, amelioration of salinity damage by PGPRs used in this study is very limited. In this type of researches, ideally, salinity stress treatment reduces plant growth and PRPRs partly recover it. This manuscript showed salinity treatment did not significantly reduce root dry weight, however, root weight was increased by PGPR inoculation under control condition. So, this paper MUST focus on only root growth promotion under control condition. I strongly recommend to revise the title and the main text to focus.
And, I think these results are interesting if people consume quinoa roots. Mostly, shoot growth and seeds yield are important rather than root growth if it is not a root crop such as carrot and radish.
Round 3
Reviewer 5 Report
In the letter, the authors claimed “plants treated with QA1or QF11 under salinity conditions clearly show significant difference compared to control (highlighted in Orange histograms)”. At a glance of Fig. 7A, yes, heights of the orange bar graph showed clear differences, however, these three data had no significant differences because these have the same letters evaluated by statistical analysis (c- with c, QA1 with bc, QF11 with abc) as described in the figure captions. This is same to other figures. Thus, serious misleading comes from the fact that statistical evaluations are totally ignored in the manuscript.
Again, evaluation of stress tolerance should be compared using the same size of plants. The authors claimed that “Plant growth enhancement observed under control conditions, is maintained under salt stress…”. Of course, large plants show better growth performance even under stress conditions, because they have much more capacity to tolerate stress, for example, they have much more vacuole capacity to compartmentalize toxic sodium. This sounds like comparison of adults and children and in most cases, of course, adults show better performance. In Fig. 7A, B, C, D, E, F, G, H, QA1 or QF11 improved one of growth parameters and these have “statistically significant differences” under control conditions. Therefore, there is no surprise Fig. 7G and 7H showed significant differences under salinity conditions.
Also, please do not ignore the fact that 400 mM NaCl is not stressful condition, especially, parameters of aerial parts showed no “statistically significant differences” between 0 and 400 mM NaCl (blue and orange bars). The effects of 400 mM NaCl on quinoa growth is very faint and stress evaluation part is not worth publishing.
As mentioned above, regrettably, I have no choice but to decline the manuscript.
